

# Tagged tracer simulations of black carbon in the Arctic: Transport, source contributions, and budget

Kohei Ikeda[1], Hiroshi Tanimoto[1], Takafumi Sugita[1], Hideharu Akiyoshi[1], Yugo Kanaya[2], Chunmao Zhu[2], Fumikazu Taketani[2]

[1]National Institute for Environmental Studies, Tsukuba, 305-8506, Japan
[2]Japan Agency for Marine-Earth Science and Technology, Yokohama, 236-0001, Japan

*Correspondence to*: Kohei Ikeda (ikeda.kohei@nies.go.jp), Hiroshi Tanimoto (tanimoto@nies.go.jp)

**Abstract.** We implemented a tagged tracer method of black carbon (BC) into a global chemistry-transport model GEOS-Chem, examined the pathways and efficiency of long-range transport from a variety of anthropogenic and biomass burning

emission sources to the Arctic, and quantified the source contributions of individual emissions. Firstly, we evaluated the simulated BC by comparing it with observations at the Arctic sites and found that the simulated seasonal variations were improved by implementing an aging parameterization and reducing the wet scavenging rate by ice clouds. For tagging BC, we added BC tracers distinguished by source types (anthropogenic and biomass burning) and regions; the global domain was divided into 16 and 27 regions for anthropogenic and biomass burning emissions, respectively. Our simulations showed that

BC emitted from Europe and Russia was transported to the Arctic mainly in the lower troposphere during winter and spring. In particular, BC transported from Russia was widely spread over the Arctic in winter and spring, leading to a dominant contribution of 62 % to the Arctic BC near the surface as the annual mean. In contrast, BC emitted from East Asia was found to be transported in the middle troposphere into the Arctic mainly over the Okhotsk Sea and East Siberia during winter and spring. We identified an important "window" area, which allowed a strong incoming of East Asian BC to the Arctic (130°–

180°E and 3–8 km altitude at 66°N). The model demonstrated that the contribution from East Asia to the Arctic had a maximum at about 5 km altitude due to uplifting during the long-range transport in early spring. The efficiency of BC transport from East Asia to the Arctic was smaller than that from other large source regions such as Europe, Russia and North America. However, the East Asian contribution was most important for BC in the middle troposphere (41 %) and BC burden over the Arctic (27 %) because of the large emissions from this region. These results suggested that the main sources

of the Arctic BC differed with altitude. The contribution of all the anthropogenic sources to Arctic BC concentrations near the surface was dominant (90 %) on an annual basis. The contributions of biomass burning in boreal regions (Siberia, Alaska and Canada) to the annual total BC deposition onto the Arctic were estimated to be 12–15 %, which became the maximum during summer.





# 1 Introduction

Arctic temperatures have increased more rapidly than the global average during the recent decades (Shindell and Faluvegi, 2009). While increases in long-lived greenhouse gases certainly play a leading role in Arctic warming, short-lived climate pollutants (SLCPs) such as aerosols and tropospheric ozone also have a substantial influence on Arctic climate (Shindell, 2007; Quinn et al., 2008; Sand et al., 2016). Black carbon (BC) has particularly attracted interest due to its large influences on radiative forcing in the Arctic (AMAP, 2015). BC causes a heating in the atmosphere by absorbing solar radiation, which is more efficient in the Arctic because of the high surface albedo of snow and ice (Quinn et al., 2007). In addition, deposition of BC on snow and ice reduces the surface albedo and results in faster-melting snow and ice sheets in the Arctic (Hansen and Nazarenko, 2004; Flanner et al., 2007). Enhanced aerosol concentrations can also increase cloud longwave emissivity and lead to surface warming in the Arctic (Lubin and Vogelmann, 2006; Garrett and Zhao, 2006). In the Arctic, air pollution and climate change are strongly linked and reductions in the concentrations of SLCPs could contribute to mitigating the Arctic warming (Quinn et al., 2008; Arnold et al., 2016).

Aerosols in the Arctic show a distinct seasonal variation with a maximum during winter and early spring and a minimum in summer (Barrie, 1986). Arctic air pollution including high concentrations of aerosols and reactive gases (so-called Arctic haze) is primarily originated from anthropogenic pollutants transported from the northern midlatitudes (Law and Stohl, 2007). The seasonal variation of the Arctic air pollution is caused by enhanced transport of pollutants from the mid-latitudes and inefficient removal processes in winter and spring and increased wet scavenging during summer (Law and Stohl, 2007; Garrett et al., 2011).

Previous studies using chemical transport models (CTMs) and chemical climate models (CCMs) revealed that these models had difficulty in reproducing the seasonal variations of aerosols in the Arctic (Shindell et al., 2008; Koch et al., 2009; Lee et al., 2013). Most models underestimated the concentration levels of BC in the peak season, and the model-to-model differences were also quite large (Shindell et al., 2008). This is caused by uncertainties in the model treatments of transformation from hydrophobic to hydrophilic BC and removal processes during the long-range transport from source regions to the Arctic. The seasonal variation of simulated BC in the Arctic is particularly sensitive to parameterizations of BC aging (Liu et al., 2011; Lund and Berntsen, 2012) and wet scavenging processes (Liu et al., 2011; Bourgeois and Bey, 2011; Browse et al., 2012). This is consistent with observational analyses by Garrett et al. (2011) who suggested that the wet scavenging process was dominant in determining the seasonal variations of light absorption and light scattering aerosols in the Arctic. Although a recent model intercomparison study indicated that the model performance of the BC simulations in the Arctic has improved, the seasonal amplitude at the surface was too weak and the BC concentration levels at the surface sites were still underestimated in the Arctic haze season in many state-of-the-science models (Eckhardt et al., 2015). These




difficulties in the model simulation of the Arctic BC are key uncertainties in calculating the source contributions from important emission sources in the northern mid- and high-latitudes.

In addition to the model representations of BC aging and removal processes, it has been recently reported that missing emission sources in the high-latitudes significantly contribute to the underestimation of simulated BC in the Arctic (Stohl et al., 2013; Huang et al., 2015). Stohl et al. (2013) estimated that gas flaring in Russia that is not treated in most inventories contributes 42% to the annual mean BC concentrations near the surface in the Arctic. Huang et al. (2015) also showed that newly developed BC emissions for Russia which includes emissions from gas flaring improved the model biases of BC at the surface sites in the Arctic.

Previous efforts of investigating the source regions of BC in the Arctic were made using a Lagrangian trajectory model (Stohl, 2006; Hirdman et al., 2010) and chemical transport models (Koch and Hansen, 2005; Shindell et al., 2008; Huang et al., 2010; Bourgeois and Bey, 2011; Wang et al., 2011; Sharma et al., 2013; Wang et al., 2014). These previous studies revealed that major BC sources transported to the Arctic were anthropogenic emissions in Europe, Russia, Asia, and North America. However, the relative importance among these source regions is still rather uncertain or even contradictory because the estimated contributions to the Arctic BC varies in earlier studies (Wang et al., 2014). For instance, while Lagrangian trajectory model analyses suggested that northern Eurasia was the major source of BC near the surface in the Arctic (Stohl, 2006; Hirdman et al., 2010), Koch and Hansen (2005) estimated that the degree of the contribution from South and East Asia was similar to that from Europe and Russia during winter and spring. In the middle troposphere over the Arctic, some studies suggested that the contributions from Europe and/or Russia were larger than or comparable to those from Asia (Shindell et al., 2008; Huang et al., 2010; Sharma et al., 2013), but other studies indicated that the contribution from Asia was dominant (Koch and Hansen, 2005; Wang et al., 2011; Wang et al., 2014). This highlights the need and importance of mechanistic understanding of transport pathways and wet removal processes during long-range transport from individual major source regions to the Arctic.

Previous studies have also reported that biomass burning emissions from boreal forests in Siberia and North America and agricultural fires in Europe have substantial influences on the Arctic BC especially from late spring to summer (Stohl et al., 2006, 2007; Warneke et al., 2010; Matsui et al., 2011). Stohl (2006) suggested that the contribution from Siberian forest fires to the Arctic was greater than that from anthropogenic sources during summer. Matsui et al. (2011) indicated that the biomass burning emissions in Russia had the most important contributions of BC in the North American Arctic in spring 2008, when severe fires occurred in Siberia. Emissions from fires in boreal forests may increase under the future warm climate (Stocks et al., 1998). Thus, it is important to investigate the contribution from biomass burning emissions at relatively high latitudes to the Arctic BC.



In this study, we investigated the long-range transport of BC from various source regions and origins to the Arctic using a global chemical transport model GEOS-Chem with a tagged tracer simulation for the past five years (2007–2011). The tagged tracer method was used to analyze detailed transport pathways and transport efficiencies of BC from individual sources to the Arctic. We identified an important geographic region, where the inflow of BC from major source regions into the Arctic occurred. This analysis also provides us with an interpretation of the seasonal variation of the Arctic BC and useful diagnostics of the model performance to understand the possible causes of model biases. We also quantitatively estimated the contributions of emissions from various sources to BC concentrations and depositions in the Arctic region.

## 2 Model description

We used the GEOS-Chem version 9-02 as a global chemical transport model (Bey et al., 2001). The GEOS-Chem is driven by assimilated meteorological data of Goddard Earth Observing System (GEOS-5) provided by the NASA Global Modelling and Assimilation Office (GMAO). The model used a horizontal resolution of $2° \times 2.5°$ with 47 vertical layers from the surface to 10 hPa. The dry deposition process in GEOS-Chem adopts a standard resistance-in-series scheme as implemented by Wang et al. (1998). Over snow and ice, BC dry deposition velocity is set to $0.03$ cm$^{-1}$ to improve aerosol concentrations at the Arctic surface sites as described in Fisher et al. (2011) and Wang et al. (2011).

## 2.1 Emission inventories

For anthropogenic emissions of BC, GEOS-Chem originally uses an inventory of Bond et al. (2007) for 2000. Wang et al. (2011) indicated that emissions in Asia and Russia were required to be doubled for matching with observed BC over the Arctic. This doubling was done to account for the emission increases since 2000 in Russia and China (Wang et al., 2011). In this study, we adopted the BC emissions of HTAPv2.2 (Janssens-Maenhout et al., 2015) which had been developed for the experiments of HTAP phase 2 for anthropogenic emissions. The target year of HTAPv2.2 was 2010 and global annual emissions were estimated to be $5.5$ Tg yr$^{-1}$, which was about 20 % larger than that of Bond et al. (2007) ($4.5$ Tg yr$^{-1}$). On a regional basis, the emissions from China were 40 % larger than those of Bond et al. (2007), and the emissions from Europe and North America were 30 % and 10 % smaller than those in Bond et al. (2007), respectively. As argued in recent studies, BC emissions from Russia may be underestimated due to missing sources such as gas flaring and have a significant impact on the Arctic BC (Stohl et al., 2013; Huang et al., 2015). Annual BC emissions in Russia were estimated to be $224$ Gg yr$^{-1}$ in Huang et al. (2015), which was about 2.5 times larger than those of HTAPv2.2. Our preliminary simulations found that the model result replacing HTAPv2.2 emission in Russia by the inventory of Huang et al. (2015) improved the reproducibility of the observed BC concentrations at the Arctic sites, and thus we used this emission dataset as the anthropogenic BC emissions. For biomass burning emissions, we used GFED (Global Fire Emissions Database) v3.1 with $0.5° \times 0.5°$ spatial resolution and daily temporal resolution (van der Werf et al., 2010). In GFEDv3.1 the BC emissions from biomass burning were globally estimated to be $1.9$ Tg yr$^{-1}$, averaged for 2007–2011.



## 2.2 BC aging and wet scavenging schemes

In the standard GEOS-Chem, 80% of BC is initially emitted as hydrophobic BC and then converted to hydrophilic BC with a constant e-folding time of 1.15 day (Park et al., 2005). However, it is unknown whether it is appropriate to adopt a constant value for the entire atmosphere. Because this value was estimated from observations of continental outflow near the source regions in the mid-latitudes (Park et al., 2005), it may be overestimated especially in remote regions including the high latitudes. In this study, we implemented a parameterization of BC aging developed by Liu et al. (2011) into GEOS-Chem and tested this impact on BC concentrations over the Arctic. This parameterization derives a time scale of BC aging based on the number concentration of OH radical (Liu et al., 2011). In remote areas including the high latitudes, the aging time is expected to be longer than that in the mid-latitudes near the source regions, resulting in an increase in BC concentrations. Liu et al. (2011) showed that the simulated seasonal variations at Arctic sites were improved by implementing this parameterization due to the increases in the BC concentrations during winter and spring.

Wet scavenging processes are also important to simulate BC in the Arctic region. The wet scavenging scheme for aerosols in GEOS-Chem is originally described by Liu et al. (2001). Wang et al. (2011) implemented several improvements for wet scavenging to distinguish between liquid and ice clouds for in-cloud scavenging (rainout) by comparing it with ARCTAS aircraft measurements over the Arctic. In liquid clouds (T≥258 K), hydrophilic aerosols are assumed to be incorporated in the cloud droplets. In the case of ice clouds (T<258 K), the model assumes that hydrophobic BC can serve as ice nuclei. However, the scavenging of BC by ice clouds is highly uncertain (Wang et al., 2011). The assumption of 100 % of hydrophobic BC can lead to overestimation of BC scavenging in ice clouds. We conducted a sensitivity simulation in which the scavenging rate of hydrophobic BC was reduced to 5% of water-soluble aerosols for liquid clouds following earlier model studies (Bourgeois and Bey, 2011). We found that the reducing scavenging rate by ice clouds improved the model reproducibility of BC at the Arctic sites in winter and spring, as will be discussed in detail below.

## 2.3 BC tracer tagging by sources and regions

In the tagged tracer simulations, we distinguished the BC tracers by source types (i.e., anthropogenic and biomass burning) as well as regions. The horizontal definitions of source regions are shown in Fig. 1. For the tagging of anthropogenic (AN) BC, we divided the global domain into 16 regions (Fig. 1a). We separated the major source regions of anthropogenic BC such as Europe, Russia, Asia and North America into different tracers. Asia was separated into three regions (i.e., East Asia, Southeast Asia and India). East Asia was further divided into four regions: Japan, Korean Peninsula, North China, and South China. For biomass burning (BB) emissions, we separated the model domain into 27 regions (Fig. 1b). For boreal forests, Siberia was separated into 6 regions based on vegetation types, and North America was divided into Alaska, West Canada and East Canada in addition to the United States.



We performed the tagged simulation for five years from 2007 to 2011 after a model spin-up for six months. The model simulation was conducted as an off-line aerosol simulation and used an improved wet scavenging and aging process. The monthly average OH distributions for the calculation of BC aging time were stored by the full-chemistry simulation for each year.

To examine the role of wet removal during transport for each tagged BC tracer, we estimated the wet scavenging ratio of BC. Using the wet scavenging ratio we discuss the differences in transport efficiency among source regions and the roles of wet removal processes for the seasonal variations of BC concentrations. We conducted an additional simulation in which the wet scavenging processes were off and thus BC was removed from the atmosphere only by dry deposition at the surface. The wet

scavenging ratio of each BC tracer was estimated as follows:

Wet scavenging ratio (%) $= (C_{wetoff} - C_{ctl})/C_{ctl} \times 100,$                              (1)

where, $C_{ctl}$ and $C_{wetoff}$ are 6-hourly BC concentrations of the control run and the simulation in which wet the removal processes are off, respectively.

## 3 Results and discussion

### 3.1 Model-observation comparison

The BC mass concentrations simulated by GEOS-Chem were compared with measurements of equivalent BC (EBC) converted from aerosol light absorption at four Arctic sites: Barrow, Alaska (156.6°W, 71.3°N, 11 m a.s.l.), Alert, Canada (62.3°W, 82.5°N, 210 m a.s.l.), Zeppelin, Norway (11.9°E, 78.9°N, 478 m a.s.l.) and Tiksi, Russia (128.9°E, 71.6°N, 8 m a.s.l.). Aerosol light absorption is observed by particle soot absorption photometers (PSAPs) at Barrow, Alert and Zeppelin,

and by an aethalometer at Tiksi. EBC is calculated from the particle light absorption coefficient with an assumption of a mass absorption efficiency. In this study, the measured light absorption coefficients with PSAPs have been converted to EBC mass concentrations using the mass absorption efficiency of 10 m$^2$ g$^{-1}$ (Bond and Bergstrom, 2006). The conversion to EBC has been internally performed by the aethalometer for Tiksi.

Figure 2 shows the seasonal variations of BC concentrations simulated with the GEOS-Chem standard scheme and our new scheme in comparison to the observations at the Arctic sites. The observed seasonal variations of BC at the Arctic surface sites show a maximum during winter and early spring (i.e., Arctic haze season) and a minimum in summer. This observed seasonal feature was relatively well simulated with the standard scheme at the semi-quantitative level (the correlation coefficients between the modeled and the observed BC ($R$) were 0.69–0.94). In contrast, the new scheme improved the

reproducibility of the model ($R$=0.81–0.94). This is mainly because the new scheme yielded an increase in BC concentrations except in summer with maximum effects in winter at the all four Arctic sites. The model reproducibility of the seasonal variations was improved, in particular, at Barrow, Alert and Tiksi. For instance, the standard scheme





underestimated BC in winter and spring at Alert and Tiksi and during spring at Barrow. These negative biases were improved by introducing an aging parameterization and reducing ice cloud scavenging. This is consistent with the results of Liu et al. (2011) and Bourgeois and Bey (2011). By introducing the aging parameterization of Liu et al. (2011), the lifetime of BC was increased due to a slower time scale of aging in the high latitudes. Reducing the wet scavenging ratio by ice

clouds also increased the lifetime of BC in the cold season. Whilst there was a substantial improvement at Barrow, Alert, and Tiksi, the observations at Zeppelin showed a reasonably good agreement with the standard simulation rather than the new simulation. The new scheme yielded nearly double BC concentrations in winter, while the observed BC concentrations were somewhat lower than those at the other three sites. Previous model studies also showed similar tendencies with larger BC concentrations in the European Arctic (i.e., at Zeppelin) than those in the North American Arctic (i.e., at Barrow and Alert)

(Sharma et al., 2013; Stohl et al., 2013; AMAP, 2015). It should be noted that the mass absorption efficiency used for the conversion from the particle absorption coefficients to the EBC concentrations has an uncertainty of at least a factor of two (AMAP, 2015). Other possible reasons include an overestimation of the emissions from Russia because of their dominant contribution to Zeppelin or a too effective transport to Zeppelin in the model.

We further compared the vertical profiles of BC concentrations over the Arctic with the observations during the Arctic Research of the Composition of the Troposphere from the Aircraft and Satellites (ARCTAS) campaign made in April 2008 (Fig. 3). Since the ARCTAS aircraft campaign covered mainly the North American Arctic, the observations made in the area north of 66°N were used. The model results by the new scheme were analyzed at the grid closest to the locations and times of the observations. The observed and simulated BC concentrations were averaged for 1-km altitude intervals from the

surface to 10 km altitude. The observed vertical profile showed a maximum in the middle troposphere at 5 km altitude. Although the model slightly underestimated the observed BC concentrations from 3 to 7 km altitude, the model successfully captured the observed mean vertical profile, including the peak in the middle troposphere as well as the concentration level near the surface.

In addition to the Arctic region, we compared the model results with measurements in the major anthropogenic source regions: East Asia, Europe, and North America. For East Asia, we used BC data at nine rural and remote sites in China during 2006 and 2007 by Zhang et al. (2012). In addition, we used measurements at Fukue Island, a remote site located in western Japan (Kanaya et al., 2016). For North America, the data from the IMPROVE network for 2007−2011 was used. In this study, we selected 43 IMPROVE sites located above 1500 m altitude for comparison. For Europe, we used

measurements at 12 sites by EUSAAR (European Supersites for Atmospheric Aerosol Research) for 2007−2011. Figure 4 shows the scatterplot of the annual mean BC concentrations simulated by the model with the new scheme in comparison to the observations in these three regions. The normalized mean bias (NMB) for East Asia was −42 %, mainly because the model largely underestimated the observations at two sites located in western China. Without these two sites, the NMB for East Asia was improved to −19 %. For Northern America, the simulated concentration levels were in good agreement with



the observations (NMB=−6 %). For Europe, the model tended to underestimate the observations (NMB=−33 %). Overall, these model-to-observations comparisons showed that our model simulations with the new scheme reasonably reproduced the observed BC levels, horizontal and vertical distributions, and spatial and temporal variabilities, thus demonstrating the model's capability to examine the long-range transport of BC to the Arctic and its underlying physical and chemical mechanisms.

## 3.2 BC transport from anthropogenic sources to the Arctic

Figure 5 shows the horizontal distributions of tagged BC tracers of major anthropogenic sources (AN) and their fluxes at about 1 km altitude in winter (DJF), spring (MAM), and summer (JJA). East Asia (EAS-AN) was defined as the sum of Japan (JPN-AN), the Korean Peninsula (KOR-AN), North China (NCH-AN) and South China (SCH-AN). North America (NAM-AN) was defined by adding Alaska and Canada (ALC-AN) to NAM-AN. BC originating from Russia (RUS-AN) widely distributed over the Arctic during winter and has a large contribution (30–100 ng m$^{-3}$) over almost the entire Arctic region. The RUS-AN contribution showed a maximum in central Siberia, which is a large source region of gas flaring (Fig. 1, Huang et al., 2015). Horizontal distributions of wet scavenging ratio are also shown in Fig. 5. The wet scavenging ratio of RUS-AN was lower than those of the other source regions especially during winter. The meteorological conditions in Russia during cold season are characterized by low precipitation and cold temperatures at the surface. These meteorological conditions lead to ineffective removal and hence effective transport from Russia to the Arctic in winter and spring. Strong northeastward fluxes from Europe (EUR-AN) were seen at 1 km altitude in winter and spring. BC originating from EUR-AN was enhanced over European Arctic during winter (20–50 ng m$^{-3}$) and spring. This result is consistent with previous studies which showed that the high-latitude Eurasia (i.e. Russia and Europe) was an important source region of BC at the surface in the Arctic (Stohl, 2006; Hirdman et al. 2010).

The horizontal fluxes of East Asia BC (EAS-AN) and North America BC (NAM-AN) showed that the long-range transport from East Asia and North America to the Arctic was inefficient in the lower troposphere. In winter, BC from East Asia was transported mainly southeastward by northwesterly winds associated with the winter monsoon circulation. BC from EAS-AN had a contribution of 10–20 ng m$^{-3}$ in the Eurasian and North American Arctic during winter and spring. The NAM-AN contribution was estimated to be 5–10 ng m$^{-3}$ in the North American Arctic during winter and spring. The long-range transport of BC from these four source regions was very weak during summer compared with the other seasons. This is because precipitation increases and wet removal becomes effective during summer.

The Horizontal distributions of tagged BC tracers and their fluxes at 5 km altitude are shown in Fig. 6, highlighting the long-range transport of BC in the middle troposphere from individual source regions. In the middle troposphere, BC originating from East Asia (EAS-AN) was transported eastward and northeastward in winter and spring. The eastward pathway from East Asia reached North America across the North Pacific. BC from East Asia also spread northeastward over the Okhotsk



Sea and East Siberia and reached the Arctic. BC from East Asia had a contribution of 20–40 ng m$^{-3}$ in the Eurasian Arctic in winter and spring. This transport pathway agreed with the results of Di Pierro et al. (2011) that analyzed aerosol export events from East Asia to the Arctic region using satellite observations. The vertical profiles of aerosol observed by the CALIOP lidar onboard CALIPSO satellite showed that the pollution plumes were transported from East Asia to the Arctic

through East Siberia in the middle troposphere (Di Pierro et al., 2011). The distribution of wet scavenging ratio showed that about 90 % of BC from East Asia was deposited before arriving at the Arctic at 5 km altitude during winter and spring. The BC transport from East Asia was much weaker in summer than those in winter and spring. BC from North America (NAM-AN) was also transported eastward and northeastward at 5 km altitude during winter and spring. In addition to eastward transport to Europe across the North Atlantic, NAM-AN BC was transported from eastern US to Greenland. The

contribution of BC from Russia (RUS-AN) in the middle troposphere was much weaker compared with the lower troposphere especially during winter (Fig. 5). The stable condition by cold temperatures near the surface suppresses the upward transport of BC over Russia especially in winter. BC from Europe (EUR-AN) at 5 km altitude was also smaller than that at 1 km altitude.

Figure 7 shows the longitude-height distributions of the meridional fluxes of BC from individual source regions at 66°N in winter, spring and summer. From these figures, we can identify important regions where inflows of BC from major source regions to the Arctic occur. A significant BC transport from EUR-AN toward the Arctic was seen at 0°–60°E below 2 km altitude in winter and spring. Transport from RUS-AN to the Arctic occurred mainly in the lower troposphere at 30°–90°E. Due to the stable condition over Russia, the inflow from RUS-AN to the Arctic was concentrated below 1 km altitude during

winter. A strong inflow from EAS-AN to the Arctic was seen in the middle-upper troposphere, and the low-level transport to the Arctic was weak in contrast to EUR-AN and RUS-AN. BC from EAS-AN was uplifted during the long-range transport to the Arctic due to the large latitudinal gradient in the potential temperature (Klonecki et al., 2003). A strong poleward transport of EAS-AN BC occurred at 130°–180°E at 3–8 km altitude during winter. Although the inflow from EAS-AN became slightly weaker than that in winter, the similar structure to winter was also seen during spring. This result was in

good agreement with the observational study by Di Pierro et al. (2011), which showed that the meridional transport of aerosol originating from East Asia to the Arctic took place at 3–7 km altitude. In summer, BC transport from EAS-AN to the Arctic was much weaker in the middle troposphere and was confined in the upper troposphere. BC transport from NAM-AN to the Arctic across 66°N was also seen in the middle-upper atmosphere, and the inflow in the lower troposphere was weak, similarly to EAS-AN. The inflow from NAM-AN to the Arctic occurred mainly at 30°–90°W at 3–8 km altitude. Pollutants

exported from East Asia and North America experience ascent transport by vertical mixing such as warm conveyer belts from the boundary layer to the free troposphere, and are eventually transported to the Arctic in the middle-upper troposphere (Klonecki et al., 2003).





The distribution of the wet scavenging ratio at 66°N showed that about 90 % of the EAS-AN BC was removed from the atmosphere during long-range transport to the Arctic in winter and spring (Fig. 7). This value is consistent with the transport efficiency (i.e., the fraction of BC not removed during transport) from Asia (13 %) derived from the BC/ΔCO ratio over the Northern American Arctic, observed during the ARCTAS spring campaign (Matsui et al., 2011). The wet scavenging ratio of

5 NAM-AN (85–90 %) was similar to that of EAS-AN. The wet scavenging ratio in the strong inflow regions of RUS-AN across 66°N (30°–90°E, below 1 km altitude) was 30–50 % during these seasons. Thus, the wet removal of the RUS-AN BC was much less than that of EAS-AN and NAM-AN, leading to an efficient transport to the Arctic. The dry condition with low precipitation in high-latitude Eurasia reduces wet deposition and leads to a longer lifetime of BC in the Arctic troposphere especially in winter. The wet scavenging ratio of EUR-AN BC at 66°N was estimated to be 40–80 % at 0°–60°E

below 2 km altitude during winter and spring.

### 3.3 Relative contributions from anthropogenic and biomass burning emissions

Figure 8 shows the seasonal variations of the individual source contributions, averaged for the Arctic (66°–90°N) from the surface to 10 km altitude. The total contribution from anthropogenic sources other than the four major source regions (Europe: EUR-AN, Russia: RUS-AN, East Asia: EAS-AN and North America: NAM-AN) was aggregated to OTH-AN. For

biomass burning (BB), the contributions from Russia (7 regions) and from Alaska and Canada (3 regions) were aggregated to SIB-BB and ALC-BB, respectively. The total contribution from biomass burning sources other than SIB-BB and ALC-BB was defined as OTH-BB. In Fig. 8, the relative contributions from individual sources to the total BC concentrations are also shown.

Due to the effective transport in the lower troposphere (Fig. 5), the contribution from RUS-AN increased from late autumn to early spring mainly below 2 km altitude. It was largest near the surface and decreased with altitude in these seasons (Fig. 8). This structure reflected a thermally stable stratification by cold temperatures at the surface during the cold season (Klonecki et al., 2003; Stohl, 2006). RUS-AN BC had a relative contribution of 40–70 % to the Arctic BC below 1 km altitude except during summer. The contribution from EUR-AN also increased below 2 km altitude in winter and early

spring, accounting for 10–20 % of the Arctic BC. EAS-AN BC increased with altitude from the surface and had the largest contribution at about 5 km altitude due to the strong poleward transport in the middle troposphere (Figs. 6 and 7). The seasonal variation of the contribution from EAS-AN showed a maximum in early spring (March) and a minimum during summer. The relative contribution from EAS-AN was estimated to be 30–50 % in the middle and upper troposphere in winter and spring. The contribution from NAM-AN showed a maximum in winter at about 5 km altitude. Because BC from

East Asia and North America located at relatively lower latitudes was emitted at higher potential temperatures, it was uplifted in the middle troposphere during long-range transport to the Arctic (Klonecki et al., 2003). OTH-AN which consisted mainly of the anthropogenic sources in the northern low latitudes and the southern hemisphere had the contribution in the upper troposphere above about 8 km altitude. In contrast to the anthropogenic sources, the contributions of biomass



burning emissions from SIB-BB and ALC-BB increased in summer because boreal fires in Siberia, Alaska and Canada increased from late spring to autumn. The relative contributions of SIB-BB and ALC-BB were estimated to be 20–40 % and 30–40 %, respectively, during summer in the lower troposphere.

Figure 9 shows the seasonal variations of the contributions from individual sources to BC mass concentrations near the surface and at about 5 km altitude averaged for the Arctic region (66°–90°N). The wet scavenging ratios of the anthropogenic sources (EUR-AN, RUS-AN, EAS-AN and NAM-AN) are also shown to highlight the role of wet removal processes on the seasonal variations of the Artic BC. Near the surface, RUS-AN was a dominant contributor of 40–70 % on a monthly basis, followed by EUR-AN (10–20 %) and EAS-AN (5–15 %) in winter, spring, and autumn. Thus, the
contributions of anthropogenic sources were remarkably larger than those of biomass burning sources during the seasons except summer. SIB-BB and ALC-BB had a substantial contribution of 10–40 % and 30–40 %, respectively, during summer, resulting in a larger contribution from biomass burning than those from anthropogenic sources in this season. At 5 km altitude, EAS-AN was the most important, accounting for 30–60 % on a monthly basis, followed by small but substantial contributions from EUR-AN (10–20 %), NAM-AN (10–15 %), RUS-AN (5–20 %), and OTH-AN (10–15 %) in winter,
spring, and autumn. The contributions of SIB-BB and ALC-BB were substantial in spring (15–20 % from SIB-BB) and summer (10–30 % from SIB-BB and 15–30 % from ALC-BB). The biomass burning contribution was comparable to that of the anthropogenic sources in summer. The relative importance to the BC concentrations on an annual basis will be discussed later (Table 2).

Near the surface, the contribution from RUS-AN showed a large seasonal variation with a maximum during winter (~100 ng m$^{-3}$) and a minimum in summer (~10 ng m$^{-3}$) (Fig. 9). BC originating from Russia was most important to the Arctic BC near the surface except during summer, and hence had a large influence on the seasonal variation of the total BC concentration over the Arctic. The wet scavenging ratio of RUS-AN had a large seasonal variation from 20 % in winter to 70 % during summer. Although the wet scavenging ratios of all four anthropogenic sources (EUR-AN, RUS-AN, EAS-AN and NAM-
AN) decreased during winter and increase in summer, the amplitude of RUS-AN was the greatest among these sources. In addition, the wet scavenging ratio of RUS-AN was the lowest among the major anthropogenic sources in all seasons, leading to a significant contribution to the Arctic BC. The seasonal variation of the contribution from EUR-AN near the surface was similar to that of RUS-AN (Figs. 8 and 9). EUR-AN was most important during winter with a contribution of ~20 ng m$^{-3}$ to the Arctic. The wet scavenging ratio of EAS-AN was the highest among the four major anthropogenic sources and exceeded
90 % in all seasons near the surface.

In the middle troposphere (at ~5 km altitude), the seasonal variation of EAS-AN BC showed an increase in spring and a decrease during summer (Figs. 8 and 9). Due to the large contribution of EAS-AN, the total BC concentration also showed a maximum in spring, which was different from the seasonal variation near the surface (winter maximum). Although the wet





scavenging ratio of EAS-AN was the largest among the major anthropogenic sources, the contribution from EAS-AN was dominant except during summer in the middle troposphere. This is because the BC emission of EAS-AN is much larger than that from the other sources as discussed below. Because EAS-AN BC was uplifted from the lower troposphere to the middle and upper troposphere during long-range transport, its contribution was larger in the middle troposphere than near the

surface. Although the wet scavenging ratio of NAM-AN was slightly less than that of EAS-AN, the contribution from NAM-AN was about 10 ng m$^{-3}$ in winter and spring and was smaller than that from EAS-AN. The contribution from RUS-AN at about 5 km altitude was much less compared with that near the surface especially in winter and spring (Figs. 8 and 9). Because of the thermally stable conditions over Russia in the cold season, the upward transport of RUS-AN BC to the middle and upper troposphere is suppressed. The contribution of EUR-AN in the middle troposphere was also smaller than

that near the surface.

**3.4 Source contributions to the annual budget of BC in the Arctic**

In Table 1, we summarized the budgets of each BC tracer averaged for 2007–2011 (see supplemental Table S1 for more detailed source regions). The annual total amount of the poleward BC flux from East Asia (EAS-AN) across 66°N which was calculated by 6-hourly concentrations and northward winds ($v>0$) was estimated to be 175.4 Gg yr$^{-1}$, corresponding to

about 10 % of the total emissions (1844.9 Gg yr$^{-1}$). The deposition amount of the EAS-AN BC on the Arctic region (66°– 90°N) was 12.3 Gg yr$^{-1}$, which was about 1 % of the EAS-AN emissions. Thus, a large part of the EAS-AN BC transported to the Arctic was transported outside of the Arctic without depositing onto the surface within the Arctic. Although the efficiency of the EAS-AN BC transport to the Arctic was lower than that of the other anthropogenic sources (EUR-AN, RUS-AN and NAM-AN) due to the effective wet removal (Fig. 9), the inflow flux was the largest among the four major

sources. This is because the emissions of EAS-AN are much larger than those from the other source regions (Table 1). On the other hand, the emissions from Russia (RUS-AN: 196.8 Gg yr$^{-1}$) were relatively small among the major anthropogenic sources, but the inflow flux was the second largest (103.0 Gg yr$^{-1}$). This is due to the effective transport from Russia to the Arctic especially during winter and spring (Figs. 5 and 9).

The global lifetimes of BC tracers were estimated to be 5.7–9.1 days (Table 1). The average lifetime of 7.3 days agreed with the value of the multi-model mean in the ACCMIP project (7.4 days, Lee et al., 2013) and with those reported by previous studies (e.g., 7.3 days from Koch and Hansen, 2005 and 5.9 days from Wang et al., 2011).

Table 2 summarized the relative contributions from individual sources to the annual mean BC concentrations, burden and

depositions over the Arctic (66°–90°N). In Table 2, the tagged BC tracers were aggregated to 5 anthropogenic and 3 biomass burning sources. As expected from Figs. 8 and 9, Russia (RUS-AN) was the most important contributor to the BC concentrations at the surface, accounting for 61.8 %. Europe (EUR-AN) had the second largest contribution at the surface (13.4 %) among the sources. The relative contribution from East Asia (EAS-AN) was estimated to be 8.0 %. This result is




similar to previous studies which showed that Northern Eurasia (Europe and Russia) was the dominant source region and East Asia had a smaller contribution at the Arctic surface (Shindell et al., 2008; Hirdman et al., 2010; Sharma et al., 2013; Wang et al., 2014). The larger contribution from Russia than Europe in this study is consistent with recent studies using newly developed emissions including gas flaring (Stohl et al., 2013; Huang et al., 2015). The contributions from biomass

burning in Siberia (SIB-BB) and Alaska and Canada (ALC-BB) were about 5 % at the surface. Thus, the contribution of anthropogenic emissions was dominant at the surface over the Arctic, accounting for 90 % in annual mean.

In the middle troposphere (5 km altitude), East Asia (EAS-AN) had the largest contribution of 40.6 % to the annual mean BC concentration over the Arctic. Among the source regions in East Asia, North China (NCH-AN) had the most significant

contribution of 29.4 % (see, supplemental Table S2). The dominance from East Asia in the middle troposphere is consistent with previous studies (Wang et al., 2011; Wang et al., 2014). The relative contribution from RUS-AN was 9.8 % at 5 km altitude, which was much less than that at the surface (62 %). Thus, the main contributor to the Arctic BC differed with altitude. This is because the transport pathways from individual sources to the Arctic are different as described before (Figs. 5–7). The transport from East Asia to the Arctic was characterized by uplifting to the middle and upper troposphere during

the long-range transport (Figs. 6 and 7). BC from Russia was transported to the Arctic mainly in the lower troposphere due to the stable condition during the cold season (Figs. 5 and 7). In the context of air pollution over the Arctic, BC from Russia and Europe is more important due to the large contributions near the surface during the Arctic haze season. In addition, BC in the lower troposphere effectively warms the Arctic surface (Flanner, 2013). On the other hand, BC in the middle troposphere is more important to radiative forcing at the top of the atmosphere and causes atmospheric heating in the lower

and middle troposphere (Flanner, 2013). Thus, it is important to understand altitudinally varying source contributions of the Arctic BC because the Arctic climate response is sensitive to the vertical distribution of BC in the Arctic.

For the BC burden over the Arctic, the contribution from East Asia (EAS-AN) was the most important and accounted for 27.4 % in annual mean. The second largest was the contribution from Russia (21.0 %). This result is consistent with AMAP

(2015) that showed that the main contributors to the BC burden in the Arctic were East and South Asia and Russia. Wang et al. (2014) also estimated that East Asia and Northern Asia (consisting mainly of Russia) had the two largest contributions of 23.4 % and 22.6 %, respectively, to the BC burden in the Arctic, consistent with this study. Bourgeois and Bey (2011) showed that Siberia, Asia and Europe had comparable contributions to the Arctic BC burden. In this study, other anthropogenic sources (OTH-AN) also had a significant contribution of 17.0 %. In OTH-AN, India (IND-AN) provided the

most important contribution of 8.7 % (see, supplemental Table S2).

We also quantitatively estimated the relative contributions to the total deposition of BC to the Arctic region (Table 2). The contribution from Russia (RUS-AN) was the largest (34.7 %). The second largest was the contribution from EUR-AN (19.0 %). Thus, the major sources of the deposition on the Arctic were identical to the dominant contributors to the BC



concentrations at the surface. This is similar to previous studies which showed that Europe and Russia provided the two largest contributions to BC deposition to the Arctic, while East Asia contributed less to deposition than to burden (Huang et al., 2010; Bourgeois and Bey, 2011; Sharma et al 2013; Wang et al., 2014), although some studies estimated a larger contribution from Europe than from Russia (Huang et al., 2010; Sharma et al., 2013; Wang et al., 2014). The contributions of

biomass burning in Siberia (SIB-BB) and Alaska and Canada (ALC-BB) were also important, accounting for 14.7 % and 12.1 %, respectively. These values of biomass burning sources were larger than their relative contributions to BC concentrations at the surface (~5 %). This is because BC deposition is enhanced during summer due to increased precipitation, and the contributions from SIB-BB and ALC-BB to the BC concentrations become large in this season in contrast to the anthropogenic sources (Fig. 9).

**4. Conclusions**

We investigated the long-range transport of BC from various source regions and origins to the Arctic and quantified source contributions using a global chemical transport model GEOS-Chem with a tagged tracer simulation for five years (2007–2011). This study especially focused on the transport pathways from the individual source regions to the Arctic and the role of wet scavenging during long-range transport. For tagging BC, we distinguished BC tracers by source types (anthropogenic

and biomass burning) and regions; the global domain was divided into 16 and 27 regions for anthropogenic and biomass burning emissions, respectively.

We evaluated the simulated BC by comparing it with observations at surface measurement sites in the Arctic and near large source regions in the northern midlatitudes. The vertical profile of modeled BC was also compared with the observations by

the ARCTAS aircraft campaign over the Arctic. We introduced a parameterization of BC aging into GEOS-Chem and changed the wet scavenging ratio by ice cloud (T<258 K). By using these new schemes, the BC concentrations were increased at the Arctic especially in winter and spring, and the model reproducibility of the seasonal variations was improved. The model also successfully reproduced the observed mean vertical distribution of BC over the Arctic.

We revealed detailed transport pathways from the individual source regions to the Arctic and identified important regions where inflow from the individual source regions to the Arctic occurred. Our simulation showed that BC originating from Europe and Russia was transported to the Arctic mainly in the lower troposphere during winter and spring. In particular, BC transported from Russia extensively distributed over the Arctic in these seasons, leading to the dominant contribution of 62 % to the Arctic BC near the surface in annual mean. We also found that this contribution of BC from Russia had a key

role in the seasonal variation of the Arctic BC at the surface. For the Arctic air pollution near the surface, BC originating from anthropogenic sources of Russia and Europe was important due to their large contributions during the Arctic haze season.



In the middle troposphere, we found a large contribution from East Asia to the Arctic BC, which resulted from uplifting during the long-range transport. Our simulation demonstrated that BC from East Asia was transported to the Arctic mainly through the Okhotsk Sea and East Siberia during winter and spring. We identified an important region where a strong inflow from East Asia to the Arctic occurred (130°–180°E and 3–8 km altitude at 66°N). The model simulation showed that the contribution from East Asia to the Arctic had a maximum at about 5 km altitude in early spring. The efficiency of transport from East Asia to the Arctic was smaller than that from other large source regions such as Europe, Russia and North America. However, the contribution of East Asia was most important to the middle troposphere (41 %) and BC burden (27 %) over the Arctic because of large emissions from this region. These results suggest that the main source of the Arctic BC differs with altitude.

The total contribution of anthropogenic sources to the BC concentrations at the surface was dominant (about 90 %) compared with that of biomass burning in annual mean. However, for BC deposition on the Arctic, the contributions of biomass burning emissions from Siberia and Alaska and Canada that became substantial during summer were important, accounting for 15 % and 12 % in annual mean, respectively.

**Acknowledgements**

Measurement data at Arctic and EUSAAR sites were obtained at http://ebas.nilu.no. Data of IMPROVE network in US were taken from http://views.cira.colostate.edu/fed. We thank NASA for providing the ARCTAS aircraft campaign data. This research was supported by the Environmental Research and Technology Development Fund (2-1505) of the Ministry of the Environment, Japan and the Coordination Funds for Promoting AeroSpace Utilization by the Ministry of Education, Culture, Sports, Science and Technology (MEXT), Japan. We thank Dr. Satoshi Inomata (National Institute for Environmental Studies) for valuable comments and discussions. This work is a contribution to PACES (air Pollution in the Arctic: Climate Environment and Societies), an activity sponsored by the IGAC (International Global Atmospheric Chemistry) project.

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





**Table 1.** Budgets of BC from individual sources for the period of 2007–2011.

| BC tracer[a] | Emission[c], Gg yr$^{-1}$ | Poleward flux across 66°N ($v>0$), Gg yr$^{-1}$ | Burden in the Arctic (66°–90°N), Gg | Deposition to the Arctic, Gg yr$^{-1}$ | | Lifetime, days |
|---|---|---|---|---|---|---|
| | | | | Wet | Dry | |
| EUR-AN | 353.7 (2.6) | 76.1 | 0.9 | 18.2 | 4.8 | 6.4 |
| RUS-AN | 196.8 (22.2) | 103.0 | 1.5 | 26.7 | 15.2 | 9.1 |
| EAS-AN[b] | 1844.9 (0.0) | 175.4 | 1.9 | 10.4 | 1.9 | 6.4 |
| NAM-AN[b] | 342.2 (0.6) | 45.5 | 0.5 | 4.5 | 0.8 | 5.7 |
| OTH-AN[b] | 2946.9 (0.1) | 110.5 | 1.2 | 4.0 | 0.7 | 7.6 |
| SIB-BB[b] | 114.2 (4.9) | 42.5 | 0.5 | 15.5 | 2.3 | 7.9 |
| ALC-BB[b] | 64.0 (5.6) | 27.0 | 0.4 | 12.6 | 2.1 | 6.3 |
| OTH-BB[b] | 1718.3 (0.0) | 21.9 | 0.2 | 1.3 | 0.1 | 8.0 |
| Total | 7580.9 (35.9) | 601.8 | 7.1 | 93.1 | 27.9 | 7.3 |

[a]AN and BB indicate anthropogenic and biomass burning sources, respectively.

[b]EAS-AN (East Asia) is the sum of JPN-AN, KOR-AN, NCH-AN and SCH-AN; NAM-AN (North America) is the sum of

5   NAM-AN and ALC-AN; OTH-AN is the sum of anthropogenic sources other than EUR-AN, RUS-AN, EAS-AN and NAM-

AN; SIB-BB is the sum of WRU-BB, S1-BB, S2-BB, S3-BB, S4-BB, S5-BB and S6-BB; ALC-BB is the sum of ALC-BB,

WCA-BB and EAC-BB; and OTH-BB is the sum of biomass burning sources other than SIB-BB and ALC-BB.

[c]Values in brackets denote emissions from north of 66°N.



**Table 2.** Relative contributions from individual sources to the annual mean BC concentrations at the surface and 5 km altitude levels, annual deposition and burden in the Arctic (66°–90°N) (%).

| Tracer[a] | Surface | 5 km | Burden | Deposition |
|---|---|---|---|---|
| EUR-AN | 13.4 | 12.2 | 12.6 | 19.0 |
| RUS-AN | 61.8 | 9.8 | 21.0 | 34.7 |
| EAS-AN[b] | 8.0 | 40.6 | 27.4 | 10.1 |
| NAM-AN[b] | 3.1 | 10.4 | 6.9 | 4.3 |
| OTH-AN[b] | 2.9 | 10.9 | 17.0 | 3.9 |
| SIB-BB[b] | 5.2 | 8.5 | 7.0 | 14.7 |
| ALC-BB[b] | 5.2 | 4.3 | 4.9 | 12.1 |
| OTH-BB[b] | 0.4 | 3.3 | 3.2 | 1.2 |

[a]AN and BB indicate anthropogenic and biomass burning sources, respectively.

5   [b]EAS-AN (East Asia) is the sum of JPN-AN, KOR-AN, NCH-AN and SCH-AN; NAM-AN (North America) is the sum of NAM-AN and ALC-AN; OTH-AN is the sum of anthropogenic sources other than EUR-AN, RUS-AN, EAS-AN and NAM-AN; SIB-BB is the sum of WRU-BB, S1-BB, S2-BB, S3-BB, S4-BB, S5-BB and S6-BB; ALC-BB is the sum of ALC-BB, WCA-BB and EAC-BB; and OTH-BB is the sum of biomass burning sources other than SIB-BB and ALC-BB.



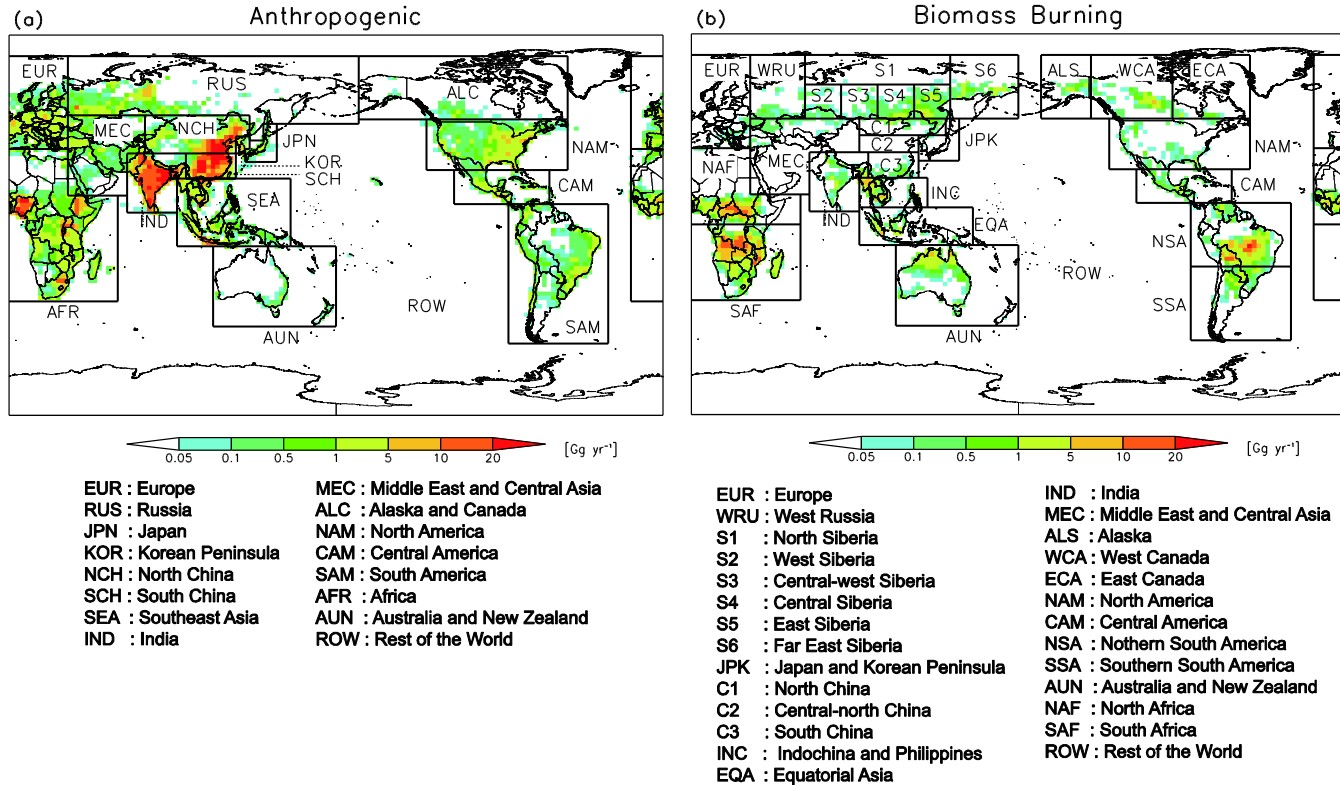

**Figure 1: Annual emissions of BC from (a) anthropogenic and (b) biomass burning sources for the year 2010 and 2007–2011, respectively, and source regions for BC tracer tagging.**





**Figure 2: Observed (black squares) and modeled (blue solid line for standard scheme and red solid line for new scheme) seasonal variations of BC mass concentrations at the Arctic sites. The plots are monthly means and the error bars are standard deviations of interannual variations. Measurements are averaged for 2007–2011 at Barrow, Alert and Zeppelin, and for 2010–2014 at Tiksi.**





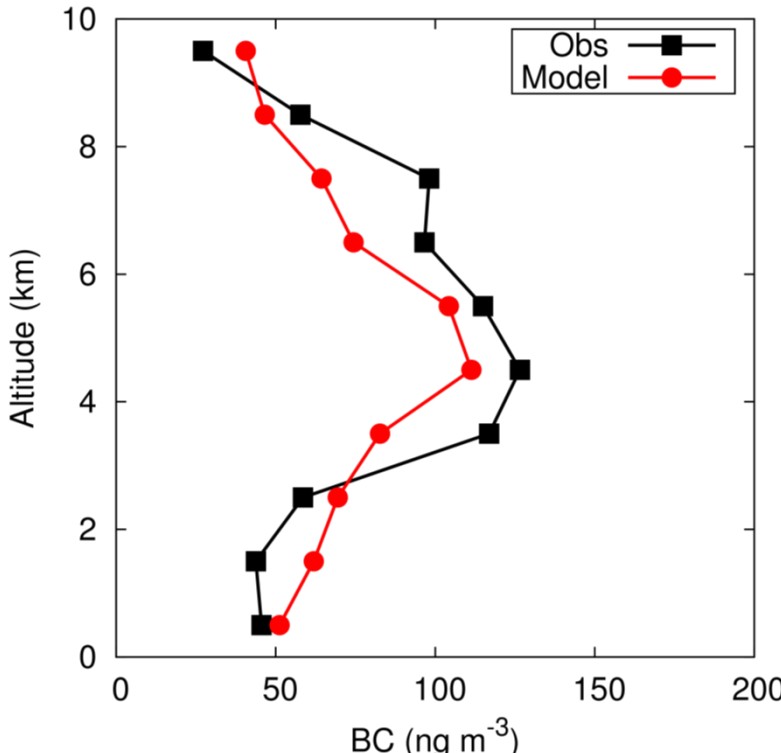

**Figure 3: Mean vertical distributions of observed and simulated BC over the region of ARCTAS aircraft campaign in April 2008.**





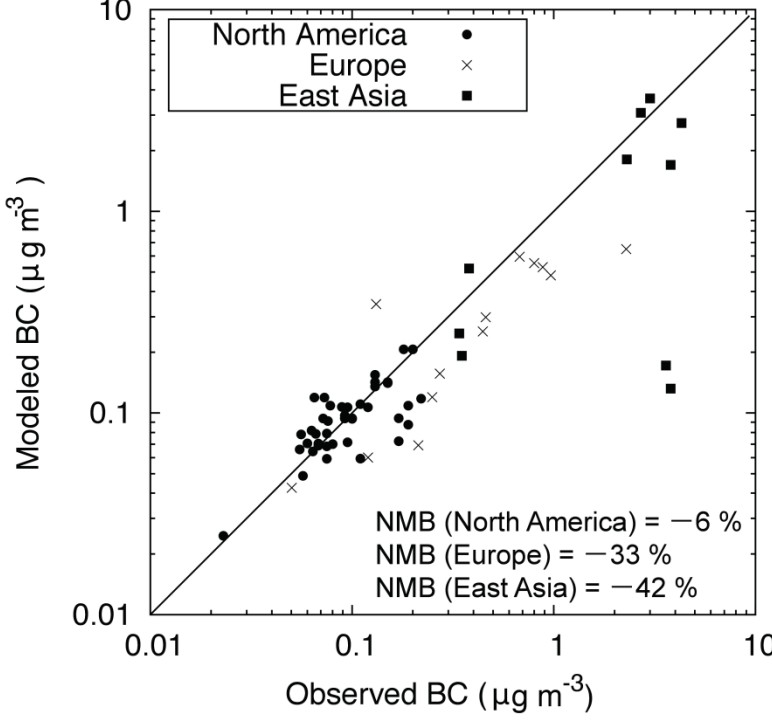

**Figure 4:** Scatterplots of annual mean BC concentrations modeled and observed at the surface sites in North America, Europe, and East Asia.



**Figure 5: Distributions of seasonal mean concentrations (color) and horizontal fluxes (arrows) at 1 km altitude for selected tagged BC tracers in winter (DJF), spring (MAM) and summer (JJA): EUR-AN, RUS-AN, EAS-AN and NAM-AN. Wet scavenging ratios are also shown by solid lines. White lines indicate the source regions of BC tracers.**





**Figure 6: Same as Fig. 5 but for 5 km altitude.**





**Figure 7: Longitude-height cross sections of mean net meridional fluxes at 66°N for selected tagged BC tracers in winter, spring and summer: EUR-AN, RUS-AN, EAS-AN and NAM-AN. Wet scavenging ratios are also shown by solid lines.**





**Figure 8: Month-altitude cross sections of mean BC concentrations from individual sources in the Arctic (66°–90°N). Relative contributions to total BC concentrations are also shown by solid lines.**





**Figure 9: Seasonal variations of mean BC concentrations (left axis) from individual sources (a) near the surface and (b) at 5 km altitude in the Arctic (66°–90°N). Mean wet scavenging ratios (right axis) for major anthropogenic source regions are also shown by solid lines: EUR-AN, RUS-AN, EAS-AN and NAM-AN.**

