# Peer review of "Tagged tracer simulations of black carbon in the Arctic: Transport, source contributions, and budget"

_Atmospheric Chemistry and Physics, 2017_

## Referee Comment (RC1) · Anonymous Referee #1 · 20 Apr 2017

This paper titled: "Tagged tracer simulations of black carbon in the Arctic: Transport, source contributions, and budget" studies the long range transport of BC from various source regions to the Arctic. The main concept of the paper, although not new, is important. However there are many issues with current version of the paper. In general several sections need be re-written. For example the authors are "barely" describing the sensitivity simulations and also there are descriptions of simulations for which no results has been shown (e.g. preliminary simulations). In addition, the literature review needs improvement with inclusion of key studies. The interpretation of comparisons between observations and model simulations are also problematic. Therefore this paper can only be recommended for publication after all of the major comments below

are addressed thoroughly and the write up is improved extensively.

Major Comments:

In abstract authors seem to claim that the new scheme has improved comparison with observations, however, this does not seem to be true when looking at figure 2 for Barrow and Zeppelin where the so called standard scheme suggests a better comparison with observation. Also why there is no blue line in Fig. 3 similar to Fig. 2?, how does blue line compare here? Why Fig. 4 does not show comparison with standard and new scheme?

Page 2, lines 30-33: I agree that Eckhardt et al. (2015) found that BC is still underestimated in several models. The potential reasons for this were investigated by Mahmood et al. (2016) who used data from same model used by Eckhardt et al., 2015 and found that one major reason is convective wet deposition process outside the Arctic which influences transport of BC into Arctic. This is a major study for Arctic BC processes and should be included in the introduction. Mahmood, R., K. von Salzen, M. Flanner, M. Sand, J. Langner, H. Wang, and L. Huang (2016), Seasonality of global and Arctic black carbon processes in the Arctic Monitoring and Assessment Programme models, J. Geophys. Res. Atmos., 121, doi:10.1002/2016JD024849

Page 4: Emission inventory: I wondering why the authors are using an older version of GFED fire emission data when a new version (GFED4 and GFED4s) are available?

Page 4, lines 10-25: It is not quite clear which anthropogenic emission inventory the authors are using. At beginning they say that "In this study, we adopted the BC emissions of HTAPv2.2", however later on they mention that they used an inventory by Huang et al., 2015. In addition, the authors also claim, without any proof, that the inventory of Huang et al., 2015 improved comparison with observation. I do not see any such results of their so called "preliminary simulations".

Page 4, lines 15-20: The doubling of BC emissions in Asia and Russia, How realistic

that would be? The authors argue that it is necessary to match modeled BC with observations in Arctic, but could this not be due to other modelling errors/discrepancy? How certain the authors are about this? A recent study showed that the differences in modeled aerosol processes in different models can contribute to overall concentrations in the Arctic (Mahmood, R., K. von Salzen, M. Flanner, M. Sand, J. Langner, H. Wang, and L. Huang (2016), Seasonality of global and Arctic black carbon processes in the Arctic Monitoring and Assessment Programme models, J. Geophys. Res. Atmos., 121, doi:10.1002/2016JD024849.)

Page 4 lines 20-22: "which was about 20%", Is 20% correct? It seems to be ~22.2% from the numbers given in that line? Even after rounding it would be 22%? Please also check the subsequent numbers.

Page 5 paragraph 25: the authors say "We separated the major source regions of anthropogenic BC such as Europe, Russia, Asia and North America into different tracers", which different tracers?

Page 5, lines 25-30: "Asia was separated into three regions (i.e., East Asia, Southeast Asia and India)", According to Fig. 1, the region named "India" contains several other countries, e.g. Sri Lanka, Pakistan, Nepal, Bangladesh, Myanmar, so this region should be named "South Asia : SA".

Page 6, lines 25-30: The authors claim that the correlations between observed and model BC has improved with new scheme. However, at least from Figure 2, it does not seem to appear that the new curve changed in its shape compared to standard scheme, only the magnitude seems to have changed then how the correlation is improved?

Page 6, lines 30-35: the authors say: " This is mainly because the new scheme yielded an increase in BC concentrations except in summer with maximum effects in winter at the all four Arctic sites.". Figure 2 clearly shows that BC is also increased in summer, though relatively small. Thus I think this sentence is not correct. Similarly from Figure

2, I do not think that the new scheme improved BC values at Barrow as the authors seem to claim. There is clearly way more over-estimations for 9 months in new scheme than the standard scheme. How can the authors claim it an improvement when it is overestimating more than the standard scheme for most months of the year including, November, December, January, February?

Page 7 lines 10-15: Is there any evidence for overestimation of BC emission from Russia?

Page 7, lines 15-30: Why there is no discussion of standard scheme in Fig. 3? If the authors want to claim that the new scheme is better than the standard scheme then all model and observations comparisons should include results from both schemes. Same comments for next paragraphs about Figure 4.

Page 9, lines 10-15: The authors write "The stable condition by cold temperatures near the surface suppresses the upward transport of BC over Russia especially in winter", I agree that stable conditions would suppress vertical transport of BC, but it would also depend on source. For example, if source is gas flaring or forest fires then emissions could reach middle troposphere?

Page 11, lines 17-18: "The relative importance to the BC concentrations on an annual basis will be discussed later (Table 2)" This sentence does not make much sense and therefore need be rewritten.

Page 12, lines 25-27, How is the BC lifetime defined here? More importantly how this discussion is related to the current study which is primarily about regional BC processes. It would more relevant if the lifetime of BC in the Arctic is given here since this study is focused on Arctic (for a multi-model comparison of lifetimes see Mahmood et al., 2016)

Page 13 lines 32-33: the authors write "We also quantitatively estimated the relative contributions to the total deposition of BC to the Arctic region (Table 2)". relative con-

tributions of what?

Page 14: Conclusions: This section has conclusions which I would find hard to agree with. For example, the authors seem to claim that they have identified important pathways for BC transport to the Arctic. Stohl (2006) had discussed the transport pathways to the Arctic. I am not satisfied that the authors provide adequate discussions of transport pathways and how they would differ with those discussed by Stohl 2006. Again authors seem to claim that new scheme improved bc simulation in Arctic which is not obvious for at least two of the four sites for which observation data they used.

Also there is no discussion of uncertainty about BC simulation results. Using just one model simulation can have problematic results. It is advisable that the author either use nudging technology or ensemble members or both to minimize the influence natural variability or at least provide some uncertainty range.

Minor comments:

Several figures can be improved. For example for Figure 4, 5 and 6 only color bar may be used instead of repeating same color bar for individual plots.

The font size of numbers of color bars is too small.
* * *

---

## Short Comment (SC1) · 12 May 2017

The authors used a tagged tracer method to assess the transport and emission sources of BC to the Arctic. It's a very interesting study and the topic is very important. I have a few short comments.

1. Using the GEOS-Chem model, a recent study (Qi et al., 2017a) systematically analyzed the key factors controlling black carbon distributions over the Arctic, such as BC emissions, wet and dry depositions. It would be very helpful if the authors could include this reference and add some discussions on it.

Reference

[Figure]

Qi, L., Li, Q., Li, Y., and He, C.: Factors controlling black carbon distribution in the Arctic, Atmos. Chem. Phys., 17, 1037-1059, doi:10.5194/acp-17-1037-2017, 2017a.

2. The authors updated the default BC aging scheme in GEOS-Chem with the Liu et al. (2011) parameterization. However, a recent study (He et al., 2016) developed a new microphysics-based BC aging scheme in GEOS-Chem, which significantly improves BC simulations. Could the authors add some discussions on it?

Reference

He, C., Li, Q., Liou, K.-N., Qi, L., Tao, S., and Schwarz, J. P.: Microphysics-based black carbon aging in a global CTM: constraints from HIPPO observations and implications for global black carbon budget, Atmos. Chem. Phys., 16, 3077-3098, doi:10.5194/acp-16-3077-2016, 2016.

3. The authors updated the BC wet scavenging by reducing the ice cloud scavenging rate. On the other hand, BC wet scavenging in mixed-phase clouds is also very important. Qi et al. (2017b) improved the BC wet scavenging in mixed-phase clouds in GEOS-Chem by incorporating an empirical parameterization. I suggest that the authors include some discussions on this aspect.

Reference

Qi, L., Li, Q., He, C., Wang, X., and Huang, J.: Effects of Wegener-Bergeron-Findeisen Process on Global Black Carbon Distribution, Atmos. Chem. Phys., In press, 2017b.

4. For the authors' information, Qi et al. (2017c) used a GEOS-Chem adjoint model to analyze the sources of surface black carbon in the Arctic. It would be useful and informative if the authors could discuss the consistency and/or inconsistency between the present study and Qi et al. (2017c) study in terms of the analyses and/or conclusions.

References

Qi, L., Q. B. Li, D. Henze, H. L. Tseng, and C. He: Sources of Springtime Surface

[Figure]

Black Carbon in the Arctic: An Adjoint Analysis, Atmos. Chem. Phys. Discuss., doi:10.5194/acp-2016-1112, 2017c.

---

## Referee Comment (RC2) · Anonymous Referee #2 · 12 Jun 2017

This is an interesting study investigating the source Black Carbon (BC) to the Arctic using GEOS-CHEM, a global chemical transport model (CTM). Ikeda et al. used a tagged tracer method to quantify the contributions of emission source the Arctic BC. The authors also discussed the seasonality of transport pathways of BC to the Arctic. In general, the paper is well-written and easy to follow, and the literature review is cohesive and complete.

Overall, I would like to recommend this manuscript for publication; however, I have few major comments and some minor comments and suggestions, that should be addressed before the paper is published.

[Figure]

Major Comments:

In this paper, the authors claim that using the new scheme "the model reproducibility of the seasonal variations is increased" or " the simulated seasonal variations were improved".

However, based on figure 2 this claim is only correct for Alert and Tiksi sites. I believe this needs further clarifications. For example, for Zeppelin site, the above claims are not true at all and the standard scheme shows significantly better performance in capturing both values and seasonality of BC. For Barrow, the standard scheme captures the summer, fall, and winter-time BC concentration better than the new scheme and we only see the improvement in the simulations for spring. Also for Tiksi, although the new scheme values are closer to observations, they are still under predicting BC very significantly. I would recommend adding some statistical analysis and more discussion for backing up this claim.

I would highly recommend comparing the results of the new scheme vs. the standard scheme for the vertical distributions along the ARCTAS flight path. Also, did you make any comparisons for each flight? Have you checked the performance of you model for the ARCTAS flights below 66N? Finally, I would recommend adding more description on the transport mechanisms from each sector and the reasons behind the seasonality. The paper shows interesting results, but it needs more discussion on how the transport pathways change in different seasons.

Minor Comments:

Page 6, Lines 17-20: Please add a reference or citations for the observation data used for this section.

Page 6, Lines 30-31: I would recommend removing the "expect in summer" phrase from the following sentence and add further clarifications to it. "This is mainly because the new scheme yielded an increase in BC concentrations except in summer with maximum effects in winter at the all four Arctic sites." Based on figure2 the new scheme shows higher values for summer as well, but the increase is smaller than other seasons.

Page 7, Line 13: What would be the possible reasons for "a too effective transport to Zeppelin"?! I would recommend adding more clarifications on why the model overestimated BC in Zeppelin.

Page 7, Lines 15-23: What would be the possible reasons for underestimation below 3k and overestimation in mid-troposphere? Adding more discussion and statistical analysis in this section will help. Also please add the standard scheme results to this analysis and figure3.

Page 7, Line 17: Please add the dates of flights used for this analysis.

Page 7, Lines 28-30: I would recommend adding references here or in page 6-lines17-20. Please see the above comment. Also please add a map with the locations of the sites that are selected for this study.

Page 7, Lines 25-30: Adding discussion on possible reasons on why the model underestimates the observations over Europe and East Asia. Also, please add the results of new scheme vs. standard scheme. How was the performance of the standard scheme for these selected sites?

Page 8, Lines 15-30: Please add some description on how you calculated meridional fluxes for these plots.

Page 9, Lines 15-32: I would recommend adding more discussions here and summarize some previous studies on Transport pathways and why there is a strong aloft meridional flux. (For example adding more discussions on location of polar dome and relative vertical mixing in different seasons).

Page 12, Lines 17-20: I have found the following sentence very confusing. Please modify this sentence. "Although the efficiency of the EAS-AN BC transport to the Arctic

was lower than that of the other anthropogenic sources (EUR-AN, RUS-AN and NAM-AN) due to the effective wet removal (Fig. 9), the inflow flux was the largest among the four major 20 sources. "

Page 13, Line 24: The second largest what? Maybe "The second largest was the contribution" -> "The second largest contributor to the Arctic BC was"?

Page 15, Lines11:15: Please add the % contributions of BB emission from Siberia and Alaska and Canada during summer.

Figure 1-a: The plot would be easier to read if you mark the whole East Asia as well, maybe adding a zoomed map for that section to show the East Asian regions (i.e. Korean Peninsula, South China, etc.) It was difficult to locate the region of East Asia and its sub-regions in the emission plot.

Figure 3: Please add the standard scheme vs new scheme comparison with observation in Figure 3. Also, it would be nice, if you can add the error bars and NMB (or RMSE).

Figure 4: It would be great to add the locations of the observations site on a map. For example, it is not obvious which IMPROVE sites were chosen for plotting and this comparison.

Figure 5 and Figure 6: I would recommend removing wet scavenging lines from these plots or export the plots at a higher resolution. The font of these plots was very small and very hard to follow. What do the numbers in the white squares represent? The numbers are very hard to read.

Figure 8 and Figure 9: Please add a description if this is the area average concentration for the Arctic or the concentration at a specified location in the Arctic?
* * *

---

## Author Comment (AC1) · 2 Aug 2017

Reply to Referee #1

Thank you very much for carefully reading our manuscript and providing valuable suggestions. We have thoroughly revised the manuscript following the comments by the reviewers.

Major Comments: In abstract authors seem to claim that the new scheme has improved comparison with observations, however, this does not seem to be true when looking at figure 2 for Barrow and Zeppelin where the so called standard scheme suggests a

better comparison with observation. Also why there is no blue line in Fig. 3 similar to Fig. 2?, how does blue line compare here? Why Fig. 4 does not show comparison with standard and new scheme?

Answer: We agree that the new scheme did not improve the reproducibility at Barrow and Zeppelin. We have modified the expression claiming that the model reproducibility has been entirely improved by the new scheme in abstract. The statement about the model performance in abstract was modified to "Firstly, we evaluated the simulated BC by comparing it with observations at the Arctic sites and examined the sensitivity of an aging parameterization and wet scavenging rate by ice clouds" (Page1, Lines 11-12). We have added the results of the standard scheme to Fig. 3 and Fig. 4 and discussions on comparisons with standard and new schemes as described below.

Page 2, lines 30-33: I agree that Eckhardt et al. (2015) found that BC is still underestimated in several models. The potential reasons for this were investigated by Mahmood et al. (2016) who used data from same model used by Eckhardt et al., 2015 and found that one major reason is convective wet deposition process outside the Arctic which influences transport of BC into Arctic. This is a major study for Arctic BC processes and should be included in the introduction. Mahmood, R., K. von Salzen, M. Flanner, M. Sand, J. Langner, H. Wang, and L. Huang (2016), Seasonality of global and Arctic black carbon processes in the Arctic Monitoring and Assessment Programme models, J. Geophys. Res. Atmos., 121, doi:10.1002/2016JD024849

Answer: We have added the following description about the study of Mahmood et al. (2016) to the introduction section (Page 2, Line 34-Page 3, Line 2).

"Mahmood et al. (2016) pointed out that convective wet deposition outside the Arctic influenced vertical distribution and seasonal variations of BC in the Arctic by analyzing the same models used by Eckhardt et al. (2015)."

Page 4: Emission inventory: I wondering why the authors are using an older version of GFED fire emission data when a new version (GFED4 and GFED4s) are available?

Answer: We compared BC emissions of GFEDv3.1 with GFEDv4.1s for 2007-2011 (simulation period) and confirmed that the difference of global BC emissions between these inventories is 9%. For boreal forests, BC emission from Siberia (defined as SIB-BB in this study) of GFEDv4.1 is only 8% larger than that of GFEDv3.1. The emission from Alaska and Canada (ALC-BB) of GFEv4.1s is 32% smaller compared with GFEDv3.1. Therefore, we think that the main conclusion of this study will not be influenced by the difference in the version of GFED.

Page 4, lines 10-25: It is not quite clear which anthropogenic emission inventory the authors are using. At beginning they say that "In this study, we adopted the BC emissions of HTAPv2.2", however later on they mention that they used an inventory by Huang et al., 2015. In addition, the authors also claim, without any proof, that the inventory of Huang et al., 2015 improved comparison with observation. I do not see any such results of their so called "preliminary simulations".

Answer: Because the emission of Huang et al. (2015) is the regional inventory for Russia, we adopted this inventory only for Russia and HTAPv2.2 was used for all regions except Russia. We have added the results of the preliminary simulation as Fig. S1. This sentence has been modified to "Our preliminary simulations found that the model result replacing HTAPv2.2 emission in Russia by the inventory of Huang et al. (2015) improved the reproducibility of the observed BC concentrations at the Arctic sites (see, Supplemental Fig. S1), and thus we used this emission dataset as the anthropogenic BC emissions for Russia" (Page 4, Lines 27-30).

Page 4, lines 15-20: The doubling of BC emissions in Asia and Russia, How realistic that would be? The authors argue that it is necessary to match modeled BC with observations in Arctic, but could this not be due to other modelling errors/discrepancy? How certain the authors are about this? A recent study showed that the differences in modeled aerosol processes in different models can contribute to overall concentrations in the Arctic (Mahmood, R., K. von Salzen, M. Flanner, M. Sand, J. Langner, H. Wang, and L. Huang (2016), Seasonality of global and Arctic black carbon processes in

theArctic Monitoring and Assessment Programme models, J. Geophys. Res. Atmos., 121, doi:10.1002/2016JD024849.)

Answer: This description is about the previous study of Wang et al. (2011) who used the inventory of Bond et al. (2007), and not for this study. We do not make any changes from the original emission data (i.e., HTAPv2.2 and the inventory of Huang et al., 2015 for Russia).

Page 4 lines 20-22: "which was about 20%", Is 20% correct? It seems to be ~22.2%from the numbers given in that line? Even after rounding it would be 22%? Please also check the subsequent numbers.

Answer: We corrected it and subsequent numbers. This part has been modified to "The target year of HTAPv2.2 was 2010 and global annual emissions were estimated to be 5.5 Tg yr−1, which was about 22 % larger than that of Bond et al. (2007) (4.5 Tg yr−1). On a regional basis, the emissions from China were 40 % larger than those of Bond et al. (2007), and the emissions from Europe and North America were 34 % and 11 % smaller than those in Bond et al. (2007), respectively" (Page 4, Lines 21-24).

Page 5 paragraph 25: the authors say "We separated the major source regions of anthropogenic BC such as Europe, Russia, Asia and North America into different tracers", which different tracers?

Answer: We have modified this sentence to "We separated Europe, Russia, Asia and North America to examine transport patterns and contributions to the Arctic from the major source regions." (Page 5, Lines 28-29)

Page 5, lines 25-30: "Asia was separated into three regions (i.e., East Asia, Southeast Asia and India)" , According to Fig. 1, the region named "India" contains several other countries, e.g. Sri Lanka, Pakistan, Nepal, Bangladesh, Myanmar, so this region should be named "South Asia : SA".

Answer: We have changed the name of this region to "South Asia" in the text, the

figures, and the tables.

Page 6, lines 25-30: The authors claim that the correlations between observed and model BC has improved with new scheme. However, at least from Figure 2, it does not seem to appear that the new curve changed in its shape compared to standard scheme, only the magnitude seems to have changed then how the correlation is improved?

Answer: We have modified the discussion on the model performance of the seasonal variations based on correlation coefficients (R) and root mean square error (RMSE) at each Arctic site. R values were improved by the new scheme from 0.89 to 0.92 at Alert and from 0.935 to 0.944 at Tiksi, respectively. At Barrow, R was increased from 0.69 to 0.81, but RMSE was not improved by the new scheme. At Zeppelin, the standard scheme (R=0.89) showed a good agreement compared with the new simulation (R=0.83). Based on these results, the discussion about the model reproducibility has been modified to the following statement (Page 7, Lines 11-16). These statistics (R and RMSE) have also been added to Fig. 2.

"The standard scheme underestimated observed BC in winter and spring at Alert and Tiksi. The model negative biases were reduced by the new scheme in these seasons, and R values were improved from 0.89 to 0.92 at Alert and from 0.935 to 0.944 at Tiksi (Fig. 2). At Barrow, while the new simulation improved the negative biases in spring, the observed concentrations were overestimated during winter. As a result, the correlation coefficient was increased from 0.69 to 0.81, but root mean square error (RMSE) was not improved by the new scheme at Barrow. Whilst there was an improvement at Alert and Tiksi, the observations at Zeppelin showed a reasonably good agreement with the standard simulation (R=0.89) rather than the new simulation (R=0.83)."

Page 6, lines 30-35: the authors say: "This is mainly because the new scheme yielded an increase in BC concentrations except in summer with maximum effects in winter at the all four Arctic sites.". Figure 2 clearly shows that BC is also increased in summer,

though relatively small. Thus I think this sentence is not correct. Similarly from Figure 2, I do not think that the new scheme improved BC values at Barrow as the authors seem to claim. There is clearly way more over-estimations for 9 months in new scheme than the standard scheme. How can the authors claim it an improvement when it is overestimating more than the standard scheme for most months of the year including, November, December, January, February?

Answer: We have removed the phrase "except in summer" from this sentence. The following statement about the seasonal variation of sensitivities (winter maximum) was added (Page 7, Lines 9-11).

"The sensitivities by changing these parameterizations were largest in winter because wet removal by ice clouds was most important in this season and aging time scale which depends on OH number concentrations also became longer than other seasons." We have modified the statement of the model performance at Barrow as described above. Please also see our reply to the above comment.

Page 7 lines 10-15: Is there any evidence for overestimation of BC emission from Russia?

Answer: We have deleted this sentence.

Page 7, lines 15-30: Why there is no discussion of standard scheme in Fig. 3? If the authors want to claim that the new scheme is better than the standard scheme then all model and observations comparisons should include results from both schemes. Same comments for next paragraphs about Figure 4.

Answer: We have added the results of the standard scheme to Fig. 3 and Fig. 4 and discussions about the comparison between the standard and the new schemes. For vertical profiles (Fig. 3), the standard scheme underestimated the observations especially in the middle troposphere. The new scheme improved the model performance by increases BC concentrations from the surface to the upper troposphere. The following

description has been added (Page 7, Line 34-Page, 8 Line 4).

"Although the standard scheme reproduced the increase from near the surface to the middle troposphere and the decrease from 5 km to the upper troposphere, the observed concentrations were underestimated 24–42 % in the middle troposphere. The negative biases were improved by the new scheme by increasing BC concentrations 18–23 ng m$-3$ in the middle troposphere. These increases by the new scheme were caused by the longer lifetime of BC in the high latitudes as discussed above."

The sensitivities were small in the major anthropogenic source regions (Europe, East Asia, and North America), because BC aging time of the new scheme is similar to that of the standard scheme ($\sim$1 day) in the mid-latitudes and wet scavenging by ice clouds is not so important in these regions. We have added the following description (Page 8, Lines 24-27).

"The differences between the standard and new schemes were small in the all three regions (Fig. 4). This is because BC aging time by the new scheme is similar to that of the standard scheme ($\sim$1 day) around the source regions in the mid-latitudes and wet scavenging by ice clouds is not so important in these regions. Because the BC concentrations tended to slightly increase in the new simulation, NMB were improved by the new scheme from $-14$–$-43$ % to $-6$–$-42$ % (Fig. 4)."

Page 9, lines 10-15: The authors write "The stable condition by cold temperatures near the surface suppresses the upward transport of BC over Russia especially in winter", I agree that stable conditions would suppress vertical transport of BC, but it would also depend on source. For example, if source is gas flaring or forest fires then emissions could reach middle troposphere?

Answer: Because fires in boreal forests occur from late spring to autumn, BC emitted from forest fires is not included in the discussion on the transport process in the cold season. Injection heights of anthropogenic sources including gas flaring are not provided by emission inventories. Thus, it is difficult to investigate the source dependence

of vertical distribution at the present stage. We would like to make this issue to address in the future research.

Page 11, lines 17-18: "The relative importance to the BC concentrations on an annual basis will be discussed later (Table 2)" This sentence does not make much sense and therefore need be rewritten.

Answer: We have removed this sentence.

Page 12, lines 25-27, How is the BC lifetime defined here? More importantly how this discussion is related to the current study which is primarily about regional BC processes. It would more relevant if the lifetime of BC in the Arctic is given here since this study is focused on Arctic (for a multi-model comparison of lifetimes see Mahmood et al., 2016)

Answer: The BC lifetime is defined as the BC burden divided by the annual total (wet and dry) deposition. The definition of BC lifetime was added to the text (Page 14, Line 13). We have added the BC lifetimes in the Arctic to Table 1 and Table S1 and a comparison with the lifetimes reported by Mahmood et al. (2016). The following statement was added (Page 14, Lines 16-21).

"The BC lifetimes of each tracer in the Arctic (66°–90°N) were estimated to be 8.6–92.7 days. The lifetime of EAS-AN BC in the Arctic (57.5 days) was longer than those of EUR-AN (14.2 days) and RUS-AN (12.9 days), because East Asia BC was distributed mainly in the middle troposphere (Fig. 9) and its deposition to the Arctic was smaller than those of EUR-AN and RUS-AN (Table 1). The average lifetime of 21.3 days in the Arctic was close to 20.0 days of the multi-model mean in the AMAP (Arctic Monitoring and Assessment Programme) models (Mahmood et al., 2016)."

Page 13 lines 32-33: the authors write "We also quantitatively estimated the relative contributions to the total deposition of BC to the Arctic region (Table 2)". relative contributions of what?

Answer: We modified this sentence to "We also quantitatively estimated the relative contributions of each source to the total deposition of BC to the Arctic region (Table 2)" (Page 15, Line 26).

Page 14: Conclusions: This section has conclusions which I would find hard to agree with. For example, the authors seem to claim that they have identified important pathways for BC transport to the Arctic. Stohl (2006) had discussed the transport pathways to the Arctic. I am not satisfied that the authors provide adequate discussions of transport pathways and how they would differ with those discussed by Stohl 2006. Again authors seem to claim that new scheme improved bc simulation in Arctic which is not obvious for at least two of the four sites for which observation data they used. Also there is no discussion of uncertainty about BC simulation results. Using just one model simulation can have problematic results. It is advisable that the author either use nudging technology or ensemble members or both to minimize the influence natural variability or at least provide some uncertainty range.

Answer: We have added discussions on transport pathways from individual sources and their seasonal variations in section 3.2. A new figure of horizontal winds in the lower and middle troposphere and precipitation was also added as Figure 7. In conclusions, we have modified the expression on transport pathways as follows. (Page 16, Lines 31-32) "We examined detailed transport pathways from the individual source regions to the Arctic and identified important regions where inflow from the individual source regions to the Arctic occurred."

We agree that the new scheme did not improve the reproducibility at Barrow and Zeppelin as replied above. We modified the expression claiming that the model reproducibility has been entirely improved by the new scheme in conclusions. The statement on the model performance in conclusions was modified as follows. (Page 16, Lines 20-24) "We introduced a parameterization of BC aging into GEOS-Chem and changed the wet scavenging ratio by ice cloud (T<258 K) to examine the sensitivities of these processes to the Arctic BC. By using these new schemes, the BC concentrations were

increased at the Arctic especially in winter and spring. Although the new scheme over-estimated the observations at Zeppelin and Barrow especially during winter, model the negative biases in the cold season were improved at Alert and Tiksi."

Because the model used in this study (GEOS-Chem) is a chemical transport model (not a chemical climate model), meteorological fields are not calculated in the model and assimilated meteorological fields GEOS-5 are used to drive it. Thus, nudging and ensemble simulations are not required in this study. We agree that it is important to provide uncertainty range. We have added the interannual variations of the source contributions to annual mean BC concentrations at the surface and 5 km altitude, annual deposition and burden in the Arctic (Table S3). Because the anthropogenic emissions used in this study had no interannual trends, interannual variations in source contributions were caused by those in meteorological conditions and biomass burning emissions. We found that results of each year were similar to that of the 5-year averaged contributions. The following description of the interannual variations of the relative contributions from each source to the Arctic BC was added in section 3.4. (Page 16, Lines 5-9)

"We estimated interannual variations of relative contributions from individual sources to the Arctic BC and found that results of each year were similar to that of the 5-year averaged contributions (see, supplemental Table S3). The differences of the relative contributions from each source to the BC concentrations between maxima and minima were lower than 12 %. For BC total deposition, the relative contribution from biomass burning in Siberia (SIB-BB) showed the variation from 8.2 % to 24.0 % (Table S3)."

Minor comments: Several figures can be improved. For example for Figure 4, 5 and 6 only color bar may be used instead of repeating same color bar for individual plots. The font size of numbers of color bars is too small.

Answer: We have modified these figures, according to the referee comment.

---

## Author Comment (AC2) · 2 Aug 2017

Reply to Referee #2

Thank you very much for carefully reading our manuscript and providing valuable suggestions. We have thoroughly revised the manuscript following the comments by the reviewers.

Major Comments: In this paper, the authors claim that using the new scheme "the model reproducibility of the seasonal variations is increased" or " the simulated seasonal variations were improved". However, based on figure 2 this claim is only correct

for Alert and Tiksi sites. I believe this needs further clarifications. For example, for Zeppelin site, the above claims are not true at all and the standard scheme shows significantly better performance in capturing both values and seasonality of BC. For Barrow, the standard scheme captures the summer, fall, and winter-time BC concentration better than the new scheme and we only see the improvement in the simulations for spring. Also for Tiksi, although the new scheme values are closer to observations, they are still under predicting BC very significantly. I would recommend adding some statistical analysis and more discussion for backing up this claim.

Answer: We agree that the new scheme did not improve the reproducibility at Barrow and Zeppelin. We have modified the expression claiming that the model reproducibility has been entirely improved by the new scheme in abstract and conclusions.

In section 3.1, we have modified the discussion on the model performance of seasonal variations based on the correlation coefficients (R) and root mean square error (RMSE) at each Arctic site. R values were improved by the new scheme from 0.89 to 0.92 at Alert and from 0.935 to 0.944 at Tiksi, respectively. At Barrow, R was increased from 0.69 to 0.81, but RMSE was not improved by the new scheme. At Zeppelin, the standard scheme (R=0.89) showed a good agreement compared with the new simulation (R=0.83). Based on these results, the discussion about the model reproducibility has been modified to the following statement (Page 7, Lines 11-16). These statistics (R and RMSE) have also been added to Fig. 2.

"The standard scheme underestimated observed BC in winter and spring at Alert and Tiksi. The model negative biases were reduced by the new scheme in these seasons, and R values were improved from 0.89 to 0.92 at Alert and from 0.935 to 0.944 at Tiksi (Fig. 2). At Barrow, while the new simulation improved the negative biases in spring, the observed concentrations were overestimated during winter. As a result, the correlation coefficient was increased from 0.69 to 0.81, but root mean square error (RMSE) was not improved by the new scheme at Barrow. Whilst there was an improvement at Alert and Tiksi, the observations at Zeppelin showed a reasonably good agreement

with the standard simulation (R=0.89) rather than the new simulation (R=0.83)."

I would highly recommend comparing the results of the new scheme vs. the standard scheme for the vertical distributions along the ARCTAS flight path. Also, did you make any comparisons for each flight? Have you checked the performance of you model for the ARCTAS flights below 66N?

Answer: We have added the results of the standard scheme to Fig. 3 and discussions about differences between the standard and the new schemes. The standard scheme underestimated the observations especially in the middle troposphere. The new scheme improved the model performance by increases BC concentrations from the surface to the upper troposphere. The following description has been added (Page 7, Line 32-Page 8, Line 2). This comparison showed averaged vertical distributions of five flights in April and did not include the observations below 66N.

"Although the standard scheme reproduced the increase from near the surface to the middle troposphere and the decrease from 5 km to the upper troposphere, the observed concentrations were underestimated 24–42 % in the middle troposphere. The negative biases were improved by the new scheme by increasing BC concentrations 18–23 ng m−3 in the middle troposphere. These increases by the new scheme were probably caused by the longer lifetime of BC in the high latitudes as discussed above."

Finally, I would recommend adding more description on the transport mechanisms from each sector and the reasons behind the seasonality. The paper shows interesting results, but it needs more discussion on how the transport pathways change in different seasons.

Answer: We added the description on the transport patterns from each source and their seasonal variations to section 3.2 based on meteorological fields. A new figure of horizontal winds in the lower and middle troposphere and precipitation was added as Figure 7. For the low-level transport from Europe and Russia, northeastward winds prevailing over northern Europe and western Russia probably played an important role

on the poleward transport in winter and spring. Low precipitation (< 1 mm day$-1$) over Russia also contributed to the effective transport to the Arctic from northern Eurasia in the cold season. In contrast, during summer the circulation pattern changed to southeastward winds and was not preferable for the poleward transport. In addition, precipitation increased over high-latitude Eurasia in summer leading to effective wet removal. The weak transport to the Arctic from Europe and Russia in summer was attributed to these meteorological conditions. The poleward transport from East Asia in the middle troposphere was attributed to northward winds blowing over the Okhotsk Sea, East Siberia and the Bering Sea in winter. Although seasonal mean northward winds in spring over these regions were weaker than those in winter, the contribution of East Asia BC in spring was larger than that in winter. This enhancement of EAS-AN BC during spring was not sufficiently explained by only the seasonal mean winds, suggesting that synoptic-scale disturbances on shorter time scales had an important role on the poleward transport from East Asia to the Arctic. Based on these results, we have added the discussions on seasonal variations of transport patterns from individual sources to the first three paragraphs in section 3.2 (Page 8 Line 31-Page10 Line 22).

Minor Comments: Page 6, Lines 17-20: Please add a reference or citations for the observation data used for this section.

Answer: We have added a reference for the observation data (Page 6, Lines 23-24). "The measurement data at the Arctic sites were obtained from EMEP and WDCA database (http://ebas.nilu.no)."

Page 6, Lines 30-31: I would recommend removing the "expect in summer" phrase from the following sentence and add further clarifications to it. "This is mainly because the new scheme yielded an increase in BC concentrations except in summer with maximum effects in winter at the all four Arctic sites." Based on figure2 the new scheme shows higher values for summer as well, but the increase is smaller than other seasons.

Answer: We have removed "except in summer" from this sentence. The following statement was added to discuss the seasonal difference of the sensitivity (Page 7, Lines 9-11). "The sensitivities by changing these parameterizations were the largest in winter because wet removal by ice clouds was most important in this season and aging time scale which depends on OH number concentrations also became longer than other seasons."

Page 7, Line 13: What would be the possible reasons for "a too effective transport to Zeppelin"?! I would recommend adding more clarifications on why the model overestimated BC in Zeppelin.

Answer: We have removed this sentence. Our simulations suggested that it is difficult to reproduce the seasonal variations at the all Arctic sites in the current model. Although the cause of the discrepancies remains unclear, it is useful to show the sensitivities of aging and wet removal by ice clouds processes at the Arctic sites. One reason is that the sensitivities of these processes at Zeppelin were larger than those at Barrow and Alert, leading to the overestimation of the new scheme in winter and spring. (Page 7, Lines 19-20)

Page 7, Lines 15-23: What would be the possible reasons for underestimation below 3k and overestimation in mid-troposphere? Adding more discussion and statistical analysis in this section will help. Also please add the standard scheme results to this analysis and figure3.

Answer: We have added the result of the standard scheme and discussion as described above. The possible reason was added (Page 8, Lines 7-8). "The simulated vertical gradient from the surface to the middle troposphere was slightly weaker than that of the observations. One possible reason is that upward transport of BC was underestimated by the model."

Page 7, Line 17: Please add the dates of flights used for this analysis.

Answer: We have added the dates of flights used for this analysis (Page 7, Line 31). "The dates of flights used for the comparison were April 8, 9, 12, 16, and 17."

Page 7, Lines 28-30: I would recommend adding references here or in page 6-lines17-20. Please see the above comment. Also please add a map with the locations of the sites that are selected for this study.

Answer: We have added references for IMPROVE and EUSAAR sites as follows (Page 8, Lines 14 and 16). A map of the sites used in this study was also added to Figure 4. "For North America, the data from the IMPROVE network for 2007–2011 was used (http://views.cira.colostate.edu/fed)." "The measurement data at EUSAAR sites were obtained from EMEP and WDCA database (http://ebas.nilu.no)."

Page 7, Lines 25-30: Adding discussion on possible reasons on why the model underestimates the observations over Europe and East Asia. Also, please add the results of new scheme vs. standard scheme. How was the performance of the standard scheme for these selected sites?

Answer: We have added the following discussion on possible reasons over Europe and East Asia (Page 8, Lines 22-24). "The possible reasons for the model underestimation over Europe and East Asia are that BC emissions from these regions are underestimated and removals around the source regions are overestimated by the model." We have added the result of the standard scheme to Fig. 4 and discussions about differences between the standard and the new schemes. The sensitivities were small in the major anthropogenic source regions, because BC aging time of the new scheme is similar to that of the standard scheme ($\sim$1 day) in the mid-latitudes and wet scavenging by ice clouds is not so important in these regions. We have added the following statement (Page 8, Lines 24-27).

"The differences between the standard and new schemes were small in the all three regions (Fig. 4). This is because BC aging time by the new scheme is similar to that of the standard scheme ($\sim$1 day) around the source regions in the mid-latitudes and

wet scavenging by ice clouds is not so important in these regions. Because the BC concentrations tended to slightly increase in the new simulation, NMB were improved by the new scheme from $-14--43$ % to $-6--42$ % (Fig. 4)."

Page 8, Lines 15-30: Please add some description on how you calculated meridional fluxes for these plots.

Answer: We have added the following description (Page 9, Lines 3-4). "The horizontal fluxes were calculated by multiplying 6-hourly BC mass concentrations by horizontal wind speeds and were averaged for three months. "

Page 9, Lines 15-32: I would recommend adding more discussions here and summarize some previous studies on Transport pathways and why there is a strong aloft meridional flux. (For example adding more discussions on location of polar dome and relative vertical mixing in different seasons).

Answer: We have added discussion on uplifting of East Asia and North America BC during long-range transport including the influence of the polar dome. The following statements were added.

"The Arctic lower troposphere is isolated by the closed polar dome which is formed by isentropic surfaces of lower potential temperatures and pollutants cannot easily be penetrated into the Arctic from outside of the polar front (Barrie, 1986). East Asia is located at south of the polar dome and EAS-AN BC is emitted from at higher potential temperatures. As a result, the low-level transport of East Asia BC into the Arctic was weak and it was transported at higher altitudes (Klonecki et al., 2003; Stohl, 2006)." (Page 11, Lines 7-11) "This is because North America BC is also emitted from higher potential temperatures and was transported to the Arctic above the polar dome." (Page 11, Lines 14-15)

Page 12, Lines 17-20: I have found the following sentence very confusing. Please modify this sentence. "Although the efficiency of the EAS-AN BC transport to the Arctic was

lower than that of the other anthropogenic sources (EUR-AN, RUS-AN and NAMAN) due to the effective wet removal (Fig. 9), the inflow flux was the largest among the four major sources. "

Answer: We have modified this sentence to "Although the fraction of BC from East Asia transported to the Arctic was lower than those of the other anthropogenic sources (EUR-AN, RUS-AN and NAM-AN) due to the effective wet removal (Fig. 9), the inflow flux of EAS-AN was the largest among the four major sources." (Page 14, Lines 5-7)

Page 13, Line 24: The second largest what? Maybe "The second largest was the contribution" -> "The second largest contributor to the Arctic BC was"?

Answer: This sentence was modified to "The second largest contributor to the BC burden over the Arctic was Russia (21.0 %)". (Page 15, Lines 17-18)

Page 15, Lines11:15: Please add the % contributions of BB emission from Siberia and Alaska and Canada during summer.

Answer: We have added the relative contributions of BB from Siberia (32 %) and Alaska and Canada (31 %) to BC deposition on the Arctic during summer. (Page 17, Lines 19-21) "However, for BC deposition on the Arctic, the contributions of biomass burning emissions from Siberia and Alaska and Canada that became substantial during summer were important, accounting for 15 % (32 %) and 12 % (31 %) in annual mean (during summer), respectively."

Figure 1-a: The plot would be easier to read if you mark the whole East Asia as well, maybe adding a zoomed map for that section to show the East Asian regions (i.e. Korean Peninsula, South China, etc.) It was difficult to locate the region of East Asia and its sub-regions in the emission plot.

Answer: We have added a zoomed map for the East Asian region to Fig. 1(a) for clarity.

Figure 3: Please add the standard scheme vs new scheme comparison with observation in Figure 3. Also, it would be nice, if you can add the error bars and NMB (or

RMSE).

Answer: We have added the result of the standard scheme as described above. The error bars and NMB were added to Figure 3.

Figure 4: It would be great to add the locations of the observations site on a map. For example, it is not obvious which IMPROVE sites were chosen for plotting and this comparison.

Answer: We have added a map of the observation sites used in this study to Figure 4.

Figure 5 and Figure 6: I would recommend removing wet scavenging lines from these plots or export the plots at a higher resolution. The font of these plots was very small and very hard to follow. What do the numbers in the white squares represent? The numbers are very hard to read.

Answer: We modified Figure 5 and Figure 6 to high-resolution figures.

Figure 8 and Figure 9: Please add a description if this is the area average concentration for the Arctic or the concentration at a specified location in the Arctic?

Answer: We have added the averaged area (66-90N) to the top of these figures.

---

## Author Comment (AC3) · 2 Aug 2017

Reply to Short Comments #1

Thank you very much for the helpful comments.

1. Using the GEOS-Chem model, a recent study (Qi et al., 2017a) systematically analyzed the key factors controlling black carbon distributions over the Arctic, such as BC emissions, wet and dry depositions. It would be very helpful if the authors could include this reference and add some discussions on it. Reference Qi, L., Li, Q., Li, Y., and He, C.: Factors controlling black carbon distribution in the Arctic, Atmos. Chem.

[Figure]

Phys., 17, 1037-1059, doi:10.5194/acp-17-1037-2017, 2017a.

Answer: We have added the recent study of Qi et al. (2017a). (Page 2, Line 28; Page 16, Lines 27-29)

2. The authors updated the default BC aging scheme in GEOS-Chem with the Liu et al. (2011) parameterization. However, a recent study (He et al., 2016) developed a new microphysics-based BC aging scheme in GEOS-Chem, which significantly improves BC simulations. Could the authors add some discussions on it? Reference He, C., Li, Q., Liou, K.-N., Qi, L., Tao, S., and Schwarz, J. P.: Microphysics-based black carbon aging in a global CTM: constraints from HIPPO observations and implications for global black carbon budget, Atmos. Chem. Phys., 16, 3077-3098, doi:10.5194/acp-16-3077-2016, 2016.

Answer: We have added He et al. (2016) to point out the importance of microphysics-based parameterization of BC aging. (Page 2, Line 27; Page 16, Lines 27-29)

3. The authors updated the BC wet scavenging by reducing the ice cloud scavenging rate. On the other hand, BC wet scavenging in mixed-phase clouds is also very important. Qi et al. (2017b) improved the BC wet scavenging in mixed-phase clouds in GEOS-Chem by incorporating an empirical parameterization. I suggest that the authors include some discussions on this aspect. Reference Qi, L., Li, Q., He, C., Wang, X., and Huang, J.: Effects of Wegener-Bergeron-Findeisen Process on Global Black Carbon Distribution, Atmos. Chem. Phys., In press, 2017b.

Answer: We have added Qi et al. (2017b) to mention the importance of wet scavenging in mixed-phase clouds. (Page 2, Line 28; Page 16, Lines 27-29)

4. For the authors' information, Qi et al. (2017c) used a GEOS-Chem adjoint model to analyze the sources of surface black carbon in the Arctic. It would be useful and informative if the authors could discuss the consistency and/or inconsistency between the present study and Qi et al. (2017c) study in terms of the analyses and/or conclusions. References Qi, L., Q. B. Li, D. Henze, H. L. Tseng, and C. He: Sources of Springtime Surface Black Carbon in the Arctic: An Adjoint Analysis, Atmos. Chem. Phys. Discuss., doi:10.5194/acp-2016-1112, 2017c.

Answer: We have added Qi et al. (2017c) to reference (Page 3, Line 14). Qi et al. (2017c) focused on source contributions of the Arctic BC on a relatively shorter time scale (April 2008). We compared our results with previous studies that treated source contributions on seasonal and annual time scales. Thus, we would like to include the study of Qi et al. (2017c) only in the introduction section.
* * *